# Immunogenicity of Hepatitis B Vaccination in Patients with Ulcerative Colitis on Infliximab Is Attenuated Compared to Those on 5-Aminosalicylic Acid Therapies: A Prospective Observational Study

**DOI:** 10.3390/vaccines12040364

**Published:** 2024-03-27

**Authors:** Mohammad Shehab, Fatema Alrashed, Munerah Alyaseen, Zainab Safar, Tunrayo Adekunle, Ahmad Alfadhli, Talat Bessissow

**Affiliations:** 1Department of Internal Medicine, Mubarak Al-Kabeer University Hospital, Jabriya 46300, Kuwait; moshehab@moh.gov.kw (M.S.); rosethedataanalyst@gmail.com (T.A.); ahalfadhli@moh.gov.kw (A.A.); 2Department of Translational Medicine, Dasman Diabetes Institute, Dasman 15462, Kuwait; 3Department of Pharmacy Practice, College of Pharmacy, Kuwait University, Safat 13110, Kuwait; 4Division of Gastroenterology and Hepatology, Department of Medicine, McGill University Health Center, Montreal, QC H3G 1A4, Canada

**Keywords:** hepatitis B, infliximab, biologics, ulcerative colitis, IBD

## Abstract

Introduction: Hepatitis B virus (HBV) infection has been associated with chronic hepatitis and cirrhosis. Patients with inflammatory bowel disease (IBD) may be at a higher risk of HBV infection reactivation, especially those on biologic therapies. This study intends to compare the effectiveness of the HBV vaccine in patients with ulcerative colitis (UC) on infliximab (IFX) compared to those on 5-aminosalicylic acid (5-ASA). Methods: Patients with UC aged >18 years old were prospectively enrolled in the study. The patients were divided into two groups: patients treated with 5-ASA (control group) and patients treated with IFX (study group). HBV vaccination was administered (20 mcg) following the standard regimen, and Hepatitis B serum antibody (HbsAb) titers were assessed three months after the final dose. The response to HBV vaccines was categorized as an ‘adequate’ immune response (≥10 IU/L) and ‘effective’ immune response (≥100 IU/L). Results: In our final analysis of 118 patients with UC, 54.2% were male and 52.5% had extensive colitis. HBsAb titer levels were significantly higher in the 5-ASA group (126.7 ± 37.5) compared to the IFX group (55.5 ± 29.4). Stratifying HBsAb levels into two categories (≥10–99 IU/L and ≥100 IU/L) revealed a significantly greater proportion of subjects in the 5-ASA group with levels ≥100 IU/L compared to the IFX group (76.7% vs. 12.1%, *p* < 0.001). Logistic regression analysis demonstrated that patients with UC receiving 5-ASA were 23.94 times more likely to exhibit HBsAb levels ≥ 100 compared to those treated with IFX (OR = 23.94, 95% CI 8.89–64.49). Conclusion: The immune response to hepatitis B vaccination in patients with ulcerative colitis treated with IFX is attenuated compared to those treated with 5-ASA. Therefore, emphasizing the importance of HBV vaccination for patients with IBD before starting anti-TNF therapy, especially IFX, and advocating for screening is imperative in high-risk countries. Determining what levels of HBsAb provide protection and what happens to the levels over time after a booster dose are important clinical questions to be answered by follow-up studies.

## 1. Introduction

Inflammatory bowel disease (IBD) is a chronic relapsing inflammatory disorder of the gastrointestinal tract. Ulcerative colitis (UC) and Crohn’s disease (CD) are the two major types of IBD, which collectively affect approximately 6.8 million people globally [1]. Over the last decade, there has been a significant transformation in the management of IBD, with the use of targeted biological therapies replacing conventional immunosuppressant therapy. Furthermore, there is a growing trend toward using these drugs earlier and more extensively for moderate to severe disease [2].

Patients with IBD are more susceptible to infectious diseases mainly due to the treatment modalities that might be employed, like immunosuppressive agents and surgical procedures [3]. Despite advancements in IBD treatment that have led to improved mucosal healing rates, concerns remain regarding their increased risk of infectious diseases [4,5].

Hepatitis B virus (HBV) is one of the major worldwide health challenges. HBV causes public health problems partially due to its high genetic diversity, drug resistance, and immune-escape mutations in surface gene [6]. It is significantly more infectious than HIV, with a transmission rate that is 50 to 100 times higher, chronically affecting 240 million individuals worldwide [7]. Its prevalence has altered over the past 30 years because of immigration, socioeconomic progress, and the introduction of a vaccine. Research in Europe, Asia, and the Americas has identified a varied range of HBV prevalence in IBD patients, ranging from 0.5% to 15% [8]. In 2016, the World Health Organization (WHO) invited all countries to collaborate in a campaign aiming to eradicate viral hepatitis, including HBV, by the year 2030 [9].

Hepatitis B can be prevented through vaccination [10]. The most widely used schedule for hepatitis B vaccination is given at time points 0, 1, and 6 months. Although several studies have demonstrated the efficiency of HBV vaccination in healthy individuals, there have been reports of HBV infections occurring in vaccinated patients with IBD undergoing immunosuppressive and biologic therapies as well as episodes of HBV reactivation among those patients [2,11]. Additionally, the response rate to HBV vaccination in the general population is about 90%. However, the response rate to the vaccine in patients with IBD is significantly lower. Young age and vaccination during disease remission are associated with a positive response to HBV vaccination [12].

Depending on the severity of the disease, IBD is frequently treated with steroids, aminosalicylates, immunosuppressors (e.g., azathioprine/6-MP, cyclosporine, methotrexate), and biologics such as tumor necrosis factor-a (TNF) inhibitors [13]. The response rate to hepatitis B virus (HBV) vaccination in patients with IBD appears to be notably low, particularly in individuals undergoing immunosuppressive treatment or anti-TNF therapy [14,15,16,17]. A recent study on patients with IBD showed a low response to single doses of HBV vaccination in comparison with double doses [18], while another study showed that anti-TNF treatment in IBD patients has a negative impact on response to vaccination [19].

Infliximab (IFX), golimumab, and adalimumab are biologics that primarily target TNF as part of their mechanism of action. Many cases of HBV reactivation have been attributed to the administration of IFX, both alone and in combination with other IBD drugs [2,20,21]. Concerns about the reactivation of HBV infection due to immunosuppressive treatment are increasing among healthcare providers managing patients with IBD [22]. Many studies have indicated that a substantial portion of IBD patients do not achieve adequate protection through vaccination [23,24].

Previous studies showed that the immunogenicity of COVID-19 vaccines was attenuated in patients treated with IFX [25,26]. Studies concerning the effectiveness of HBV vaccination in patients with IBD taking different medications would aid in the identification of patients at high risk for loss of immune response and could contribute to the development of more explicit IBD management guidelines in the future [27]. Furthermore, a study showed that liver dysfunction due to HBV reactivation can occur in HBV-infected IBD patients treated with anti-TNF agents [28]. Additionally, there are limited data on the effectiveness of HBV vaccination in patients with IBD receiving different types of IBD medications. Thus, in this study, we aimed to assess the effectiveness of the HBV vaccine in IBD patients who were undergoing treatment with IFX or 5-aminosalicylic acid (5-ASA).

## 2. Materials and Methods

### 2.1. Study Design

A prospective observational study was conducted at Haya Alhabib Gastroenterology Center in Kuwait from January 2021 to January 2024. The Strengthening the Reporting of Observational Studies in Epidemiology (STROBE) guidelines were followed to conduct and report this study [29] (Appendix A). The 2016 International Classification of Diseases (ICD-10) was used to diagnose inflammatory bowel disease (IBD). Patients were classified as having ulcerative colitis (UC) if they displayed ICD-10 codes K51, K51.0, K51.2, K51.3, K51.5, K51.8, or K51.9 [30].

HBV seronegative patients with UC receiving 5-ASA (control group) or IFX monotherapy (study group) were eligible to be enrolled in the study. The inclusion criteria were as follows: (1) patients who received all 3 scheduled doses of HBV vaccinations; (2) adult patients (≥18 years old) who had been treated with IFX therapy or 5-ASAs for at least 6 months before the first dose of HBV. Serum HbsAB levels were measured 3 months after the last dose of HBV. The exclusion criteria were as follows: (1) pregnant women; (2) patients with active UC (defined as having any of the following: partial Mayo score of >2 with individual subscores > 1, or C-reactive protein ≥ 10 mg/L or fecal calprotectin of ≥250 ug/g); (3) patients with chronic kidney disease on dialysis; (4) patients with immune deficiency diseases such as human immunodeficiency virus (HIV); (5) patients with incomplete vaccination doses; (6) patients who stopped their IFX or 5-ASA therapy during the study period; (7) patients who were on previous biologic therapies, immunosuppressants, or steroids for less than 3 months before enrolment or concomitants use of these therapies; (8) patients who were found to be HBsAg or HBcAb positive; (9) patients with subtherapeutic doses of therapy (less than 2 g of 5-ASA per day or IFX less than 5 mg/kg I.V every 8 weeks); (10) patients who had previously received a hepatitis B vaccine.

### 2.2. Data Collection and Outcome Measure

We electronically collected all the required data from the patients, which included demographic and clinical data: age, sex, BMI, smoking status, extent of disease, IBD treatment, date of the vaccination, and serological response to the vaccine.

The enrolled patients were vaccinated against HBV using a standard regimen using 3 doses of 20 mcg hepatitis B surface antigen [HBsAg] administered at 0, 1, and 6 months (Figure 1). Patients were stratified into two separate age-, gender-, and BMI-matched groups. The control group consisted of patients treated with 5-ASA, and the study group consisted of patients treated with IFX.

Quantitative HBsAb levels were measured three months after the last dose of the vaccine, and the vaccination response was assessed. The response rate to the vaccine (immunization rate) was determined using two HBsAb serum titer cut-offs: a titer of 10 IU/L or more was considered an “adequate” immune response, while a titer of 100 IU/L was considered an “effective” immune response.

The primary outcome of the study was to investigate and compare the serological response to HBV vaccination using HbsAb level in patients with UC undergoing treatment with either IFX or 5-ASAs.

### 2.3. Statistical Analysis

Data collection was carried out using Office 365 Microsoft Excel v16.0, and statistical analysis was performed using SPSS 27.0.1.0 (SPSS, Chicago, IL, USA). Categorical variables were presented as frequencies (n, %); continuous variables were presented as mean ± standard deviation. A logistic regression was performed using the variables that provided a statistically significant association with the response to vaccination on the univariate analysis. There are no studies that directly compare the effectiveness of the HBV vaccine in UC patients treated with 5-ASA vs. IFX in the literature to calculate the sample size. Consecutively, we enrolled 60 patients in each group matched for age, gender, and BMI to ensure the appropriate representation of the population and to increase the statistical power of the study. The results were expressed as odds ratio with a 95% confidence interval (CI), and the statistical significance was set at *p* < 0.05.

### 2.4. Ethical Statement

The standing committee for the coordination of health and medical research at the Ministry of Health of Kuwait reviewed and approved the protocol of this study (protocol number 3679/2021). Our study complied with the Helsinki Declaration, and all participants gave written, informed consent to participate in the study.

## 3. Results

In total, 273 patients with ulcerative colitis treated with 5-ASA or IFX were initially screened. After applying our inclusion and exclusion criteria, it was found that two patients were HBsAg positive, and four patients were HBcAb positive. A total of 118 patients were included, with 60 individuals in the study group and 58 patients in the control group (Figure 2). Of the study participants, 54.2% were male, and the mean age of the study cohort was determined to be 34.4 ± 12.2 years. The average body mass index (BMI) for the entire study population was calculated at 25.5 ± 5.1, with 83.3% (*n* = 100) being non-smokers. Among the 118 patients included in the study, 52.5% were identified as having E3 disease (extensive colitis). In terms of laboratory values, mean (SD) albumin was 41 g/L ± (6.1) in the control group and 40 g/L ± (5.9) in the study group. CRP was 6.1 mg/L ± (4.3) in the control group and 5.8 mg/L ± (4.5) in the study group. Finally, stool fecal calprotectin was 114 ug/g ± (12.1) in the control group and 112 in the control group ± (11.9). A comprehensive summary of the baseline characteristics of the patients is presented in Table 1.

No significant differences were observed in the demographic and clinical characteristics of patients in the two groups (age-, gender-, and BMI-matched groups), except for the mean HbsAb titer levels. At 3 months post HBV vaccination, the mean HBsAb titer levels were significantly higher in the 5-ASA group when compared to the IFX group (126.7 IU/L ± 37.5 vs. 55.5 IU/L ± 29.4). On further classifying the HBsAb titer level into two groups, adequate response (≥10–99 IU/L) and effective response (≥100 IU/L), the IFX group showed a significantly higher proportion of patients with HBsAb levels ≥ 10–99 IU/L compared to the 5-ASA group (87.9% vs. 23.3%, *p* < 0.001. On the other hand, the 5-ASA group had a significantly higher proportion of patients with HBsAb levels ≥ 100 IU/L compared to the IFX group (76.7% vs. 12.1%, *p =* 0.000). The findings are presented in Appendix A.

When patients were categorized according to their response to the hepatitis B vaccine, distinguishing between an adequate response (≥10–99 IU/L) and an effective response (≥100 IU/L), no substantial differences were observed in the baseline and clinical characteristics of the study population (Appendix A).

A logistic regression was performed, and the univariate analysis showed that patients with IBD treated with 5-ASA were 23.9 times more likely to have HBsAb levels of ≥100 when compared to those treated with IFX. Additionally, the box plot shows that in patients with IBD receiving 5-ASA, HBsAb levels are higher than patients receiving IFX (Figure 3).

## 4. Discussion

The current prospective study examined the response of IBD patients undergoing treatment with IFX or 5-ASA to standard HBV vaccination regimens to assess their effectiveness. In healthy individuals, a standard dose of HBV vaccination typically results in a sufficient antibody response in more than 90% of cases, but this rate notably declines in immunosuppressed patients [31,32]. Out of the 120 IBD patients, 118 individuals were included in the analysis, meeting the specified inclusion and exclusion criteria. The study predominantly consisted of male participants, and most of the patients were observed to have extensive colitis. Mishra et al. [33] conducted a study among IBD patients showing similar demographic and clinical characteristics. Their study compared HBV vaccination response in IBD patients and controls, revealing that males had a higher response rate compared to females. In contrast, Altunöz M. E. et al. [15] conducted a study that examined vaccine response in relation to gender. The analysis found that female patients had a notably higher response to vaccination (85.18%) compared to male patients (66.6%), but the response rate was similar between patients with CD and UC (*p =* 0.302). The outcomes of these studies did not agree with our findings, as there were no significant differences observed between the two groups with respect to their demographic and clinical characteristics.

In the present study, the mean HBsAb titer levels were significantly lower in the IFX group when compared to the 5-ASA group. Similar results were observed in the study by Gisbert et al. [16]. The study aimed to evaluate the impact of immunosuppressors and anti-tumor necrosis factor (anti-TNF) agents on HBV vaccine. In total, 241 vaccinated patients with IBD were recruited, and the response rate (HBsAb) was lower in patients who were receiving anti-TNF therapy. Similarly, a study by Altunöz and colleagues displayed that IBD patients receiving several IBD-related medications had a lower response rate to HBV vaccination compared with patients without any immunosuppressive medication [15].

Belle et al. [34] evaluated the efficacy of the HBV vaccine between IBD patients and healthy controls and investigated the impact of immunosuppressive therapy on vaccine response in IBD patients between the three IBD treatment approaches (anti-TNF, thiopurine, combination therapy, and no therapy); among the 164 participants, the median titers of anti-HBs did not differ between the sub-groups, at 246.25 ± 330.88, 275.93 ± 369.99, 273.54 ± 357.58, and 306.91 ± 385.49, respectively. Vida Perez et al. [14] found that the concomitant use of immunosuppressors or biologics did not affect the efficacy of HBV vaccination. Dotan et al. [35] aimed to investigate evidence of any intrinsic systemic immunodeficiency in IBD patients. The study recruited 31 patients with CD and 12 patients with UC, and the authors found that, after 24 weeks of treatment with thiopurines at doses commonly used for managing IBD, there was no notable suppression of systemic cellular and humoral immune responses.

According to the WHO, an HBsAb concentration of >10 IU/L measured 1–3 months after the administration of the last dose of the primary vaccination is considered a reliable marker of protection against infection. The anti-HBs level declines progressively with time after a primary hepatitis B immunization schedule, and it is possible to find anti-HBs levels lower than 10 IU/L several years after vaccination [22,36]. Indeed, one study showed that over 25% of HBV vaccine recipients had an anti-HBs titer <10 IU/mL after 18 years of the primary vaccination. The Center for Disease Control (CDC) considers no responders to be patients with anti-HBs less than 10 IU/L after two complete series of Hepatitis B vaccine [37]. We classified patients based on adequate (≥10–99 IU/L) and effective response (≥100 IU/L) to HBV vaccine. The baseline and the clinical characteristics of the study population did not vary significantly. In an American population-based study, exposure to anti-TNF medications was associated with reduced antibody response in patients with IBD. Specifically, patients who had been exposed to IFX were less likely to have titer levels ≥10 (*p* < 0.01) [27]. In contrast, our study revealed that the IFX group (87.9%) had a significantly high percentage of individuals with HBsAb levels ≥10–99 IU/L (adequate response). When considering HBsAb levels of ≥100 IU/L (effective response), it is worth noting that the 5-ASA group (76.7%) exhibited a significantly higher proportion of individuals with these levels compared to the group receiving IFX (12.1%). The data on the effectiveness of HBV booster vaccination on IBD patients are limited. A single study by Chang et al. showed that out of the 44 patients IBD patients who received a booster vaccine, 13 failed to achieve an optimal vaccine response [3]. More research on the efficacy of booster vaccinations in IBD patients treated with IFX is warranted.

In the current study, it was observed that IBD patients using 5-ASA are 23.9 times more likely to have HBsAb levels of ≥100 when compared to those treated with IFX. A study by Andrade et al. [22] assessed the response rate to HBV vaccination in IBD patients. In total, 217 patients with IBD treated with IFX were recruited, and the study showed that both treatment with azathioprine (AZA) alone and combination therapy of AZA and IFX were correlated with a poor response to HBV vaccination. However, after accounting for potential confounding factors, the only predictors for a reduced response to the vaccine were the administration of azathioprine and IFX. A systematic review and meta-analysis [12] aimed to determine the response rate to HBV vaccination and to identify the factors predictive of an immune response. The reported results showed that patients without immunosuppressive therapy had a greater response to the vaccine compared with patients on immunomodulatory (RR 1.33; 95% CI 1.08–1.63) or anti-TNF therapy (RR 1.57; 95%-CI 1.19–2.08).

In the present study, even though the IFX group has a relatively lower level of HBsAb than the 5-ASA group, the HBsAb level of IFX is still above 10 IU/L. This finding has clinical significance, as patients on biologics are at a higher risk of infections and tend to have a faster decay in antibodies to vaccines over time. This has been shown in patients with IBD treated with IFX after taking a COVID-19 vaccine [38]. Additionally, it would be interesting for future studies to evaluate other relevant factors such as response to treatment, severity of the disease, and other factors.

Hepatitis B immune status is of particular importance in patients with IBD, as there are reports of reactivation of hepatitis B infection in patients starting anti-TNF therapy [16,34]. A study found that patients with IBD had significantly lower levels of HBsAb compared with healthy controls [39]. Furthermore, immunosuppressive medications of all types may lead to reactivation of HBV replication in patients with chronic HBV infection [39,40]. The HBV vaccine is effective, and the mainstay control of HBV is to prevent infection and consequent acute and chronic liver disease; however, patient adherence can be low [40,41,42]. Guidelines by the American Association for the Study of Liver Disease (AASLD) and the CDC recommend HBV screening and vaccination for any person seeking protection from hepatitis B, and also for those requiring immunosuppression (including IBD patients on steroids > 20 mg/d for 2 weeks or more, high-dose purine analogs, and other immunosuppressive agents such as TNF inhibitors) [43,44]. According to these guidelines, HBV vaccination should ideally occur before the initiation of therapy with immunosuppressants. Additionally, based on a meta-analysis, patients with IBD demonstrate a significantly inferior response to HBV, suggesting that IBD patients might benefit from an extended hepatitis B vaccination program [17].

This study has several clinical implications. Our results showed that immune response to hepatitis B vaccination in patients with UC treated with IFX is attenuated compared to those treated with 5-ASA; therefore, healthcare providers should emphasize the importance of adherence to HBV vaccination for patients with IBD before starting anti-TNF therapy. Establishing a clinical protocol engaging patients and guiding providers are some of the ways to improve adherence. Additionally, this study underlines the importance of screening for HBV infection as recommended by current guidelines and performance measures [45].

Our study has several strengths. To the best of our knowledge, this is the first study to compare IFX monotherapy with 5-ASA. It is a well-designed prospective study with an age-, sex-, and BMI-matched control arm, and it reflects real-time practice patterns at a gastroenterology tertiary center. Given our strict inclusion and exclusion criteria, potential confounders and biases are limited. It addresses an important vaccination concern, especially in countries with high prevalence. It is imperative to recognize the inherent limitations of our study. Firstly, it is a single-center study with a relatively small sample size. Furthermore, the current study has a short 9-month follow-up, which makes it difficult to determine whether the study population needs an HBV booster dose. Additionally, HBsAb titers were not checked before enrolment; therefore, we based our patient inclusion on medical records and patient history, which can be subject to recall bias. Also, it would have been interesting to have a third arm with healthy individuals to compare with patients with IBD. Moreover, our investigation only included a limited set of demographic and clinical variables, which might have made it difficult to identify all the factors associated with a successful response to HBV vaccinations. Further research comparing the impact of IFX and newer biologic drugs for IBD on the immunogenicity of HBV vaccines is warranted.

## 5. Conclusions

The immune response to hepatitis B vaccination in patients with UC treated with IFX is attenuated compared to those treated with 5-ASA. Therefore, emphasizing the importance of HBV vaccination for patients with IBD before starting anti-TNF therapy, especially IFX, and advocating for screening is imperative in high-risk countries. Determining what levels of HBsAb provide protection and how the levels are affected over time after a booster dose are important clinical questions to be answered by follow-up studies.

## Figures and Tables

**Figure 1 vaccines-12-00364-f001:**
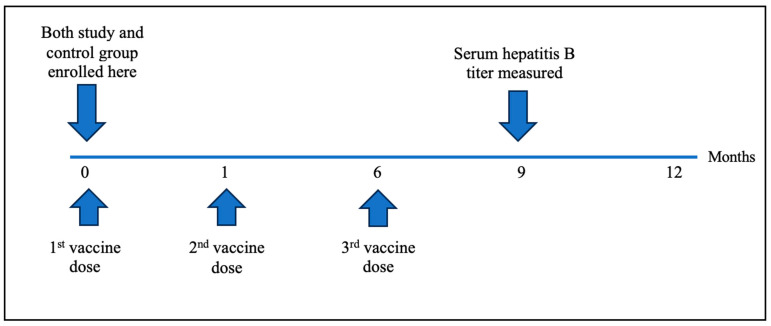
Enrolment and vaccination timeline.

**Figure 2 vaccines-12-00364-f002:**
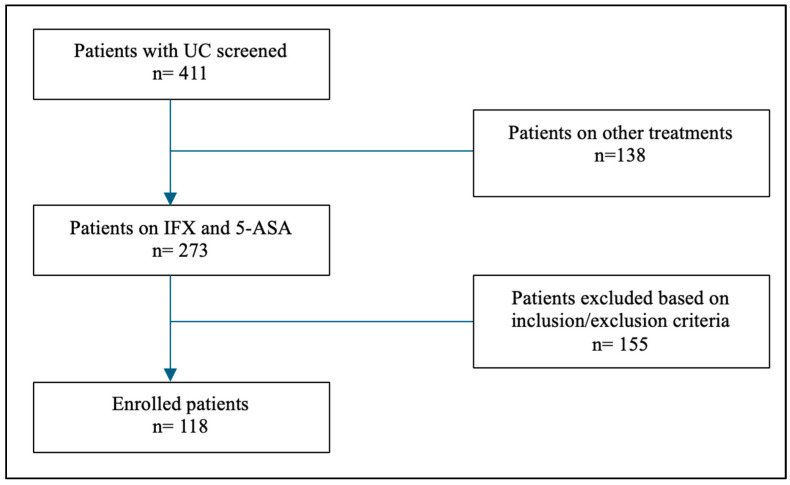
Flow chart showing study patient enrollment. IFX—infliximab; 5-ASA—5-aminosalicylate.

**Figure 3 vaccines-12-00364-f003:**
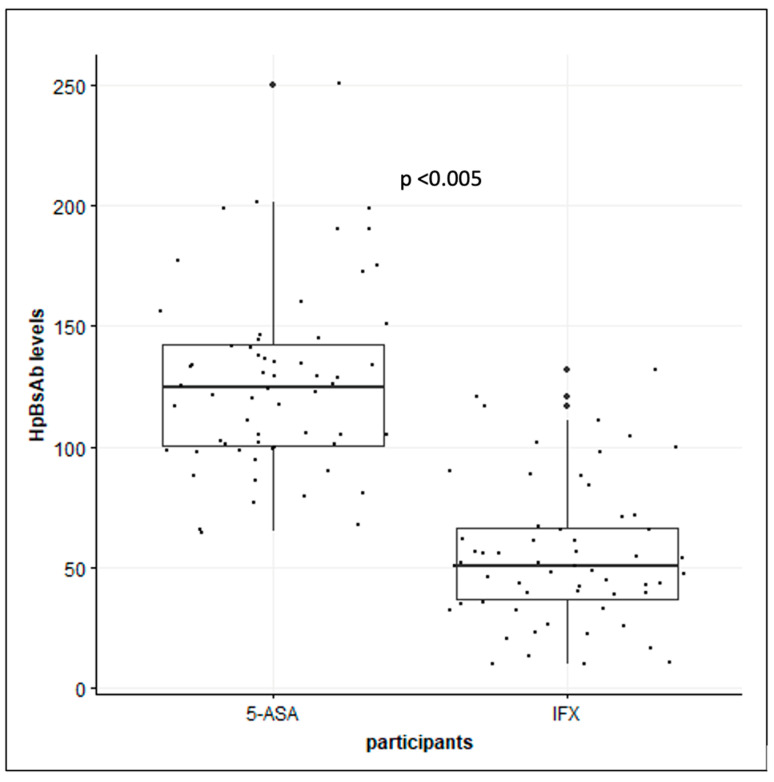
Box plot showing difference in HBsAb levels between study and control group.

**Table 1 vaccines-12-00364-t001:** Demographic baseline and clinical characteristics comparison across treatment groups.

Clinical Variables	Control Group (5-ASA)	Study Group (IFX)	*p*-Value
Age (Mean ± S.D)	34.4 ± (12.6)	34.4 ± (11.9)	0.971
BMI (Mean ± S.D)	25.6 ± (4.9)	25.5 ± (5.3)	0.917
Gender, N (%)			0.985
Male	33.0 (55.0%)	32.0 (55.2%)	
Female	27.0 (45.0%)	26.0 (44.8%)	
Smoking, N (%)			0.270
Smoker	7.0 (11.7%)	11.0 (19.0%)	
Non-Smoker	53.0 (88.3%)	47.0 (81.0%)	
UC Type, N (%)			0.935
E1	7.0 (11.7%)	7.0 (12.1%)	
E2	20.0 (33.3%)	21.0 (36.2%)	
E3	33.0 (55.0%)	30.0 (51.7%)	
Inflammatory markers (Mean ± S.D)			0.462
Albumin, g/L	41 ± (6.1)	40 ± (5.9)	
CRP, mg/L	6.1 ± (4.3)	5.8 ± (4.5)	
Stool fecal calprotectin, ug/g	114 ± (12.1)	112 ± (11.9)	
HBsAb levels (Mean ± S.D)	126.7 ± (37.5)	55.5 ± (29.4)	<0.005
HBsAb level group, N (%)			<0.005
≥10–99	14.0 (23.3%)	51.0 (87.9%)	
≥100	46.0 (76.7%)	7.0 (12.1%)	

## Data Availability

Data are available on request, from the corresponding author, due to local legal and ethical restrictions.

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
