# Peer review of "Immunogenicity of Hepatitis B Vaccination in Patients with Ulcerative Colitis on Infliximab Is Attenuated Compared to Those on 5-Aminosalicylic Acid Therapies: A Prospective Observational Study"

_vaccines, 2024, doi:10.3390/vaccines12040364_

Round 1
Reviewer 1 Report
Comments and Suggestions for Authors
Shehab et al. reported that UC patients on 5ASA are 23.9 times more likely to have anti-HBs >100 when compared to those on IFX. this is an interestin study with some flaws
1) Please add a figure with a patient’s disposition. Starting from all the patients with UC please report all the excluded according to the defined criteria. How many were anti-HBs positive?
2) Can you confirm that all the enrolled patients were anti-HBs and anti-HBc negative? None of the patients prevoiously received HBV vaccination without a response or with reduced anti-HBs levels <10 at time of enrollment?
3) I can imagine that you may have some anti-HBc positive patients. These should be excluded as considered patients with previous infection
4) Figure 1 and Table 4 are not necessary
5) Could you please tell something about HBV vaccination program in Kuwait?
6) In the discussion it is essential to discuss on the protection that all patients having anti HBs >10 may have independently of the title as well as the potential protection also in those who remained anti Hbs <10 after complete vaccination schedule
Author Response
Shehab et al. reported that UC patients on 5ASA are 23.9 times more likely to have anti-HBs >100 when compared to those on IFX. this is an interesting study with some flaws.
1) Please add a figure with a patient’s disposition. Starting from all the patients with UC please report all the excluded according to the defined criteria. How many were anti-HBs positive?
Thank you for your comment. Added to results section.
2) Can you confirm that all the enrolled patients were anti-HBs and anti-HBc negative? None of the patients previously received HBV vaccination without a response or with reduced anti-HBs levels <10 at time of enrollment?
Thank you for your comment. See added method and results section. Patients with HBsAg and HBcAb positive were excluded. In addition, patients with previously history of hepatitis B vaccination were excluded, however, HBsAb titers were not checked before enrollment.
3) I can imagine that you may have some anti-HBc positive patients. These should be excluded as considered patients with previous infection
Thank you for your comment. See added method and results section. Patients with HBsAg and HBcAb positive were excluded.
4) Figure 1 and Table 4 are not necessary
Thank you for your comment. We have removed table 4 and added to the supplementary file. As for figure 1, we feel it is an important visual depiction of the study timeline for the readers.
5) Could you please tell something about HBV vaccination program in Kuwait?
Thank you for your comment. Hepatitis vaccination is mandatory for all high-risk occupations such as health care worker. It has been also introduced in the last 10 years to be given at birth for all infants which does not include our study patient population because they are all over 18 years of age.
6) In the discussion it is essential to discuss on the protection that all patients having anti HBs >10 may have independently of the title as well as the potential protection also in those who remained anti Hbs <10 after complete vaccination schedule
Thank you for your comment. We have added this to the discussion.
Reviewer 2 Report
Comments and Suggestions for Authors
The authors aimed to present the results of an observational study on hepatitis B vaccination in UC patients with infliximab and 5-ASA treatment.
The study has some issues:
- no patients were analyzed if it was previously vaccinated. The authors are sure that nobody was previously vaccinated.
- the editing must be verified; the abbreviated words should be consistently used;
- in the abstract, the presentation of the results is doubled: lines 28-29 and lines 29-30 - inverse results (100% minus the previous). There's no need for that.
- the presentation of the results should be limited: there is no need to present the same data in 2 tables: some in Table 1 and Table 2 and others in Table 2 and Table 4. Where is table 3?
Data from the table should not be presented again as a figure. There's no need for that.
- The authors must extend the study by analyzing other factors involved in the different results, such as response to treatment, severity of the disease, and other issues.
Comments on the Quality of English Language
The paper needs editing. The English language seems to be used appropriately.
Author Response
- no patients were analyzed if it was previously vaccinated. The authors are sure that nobody was previously vaccinated.
Thank you for your comment. See added method and results section. Patients with HBsAg and HBcAb positive were excluded. In addition, patients with previously history of hepatitis B vaccination were excluded, however, HBsAb titers were not checked before enrollment. We have also added this as limitation of the study.
- the editing must be verified; the abbreviated words should be consistently used;
Thank you for your comment. We have revised the manuscript for typos and consistence abbreviations.
- in the abstract, the presentation of the results is doubled: lines 28-29 and lines 29-30 - inverse results (100% minus the previous). There's no need for that.
Thank you for your comment. We have removed the double (repeated) results.
- the presentation of the results should be limited: there is no need to present the same data in 2 tables: some in Table 1 and Table 2 and others in Table 2 and Table 4. Where is table 3?
Thank you for your comment. We have merged tables 1 and 2. And moved table 3 to supplementary material.
Data from the table should not be presented again as a figure. There's no need for that.
Thank you for your comment. We have moved table 3 to supplementary material.
- The authors must extend the study by analyzing other factors involved in the different results, such as response to treatment, severity of the disease, and other issues.
Thank you for your comment. As per our inclusion and exclusion criteria, any patients with active disease, who were not responding treatment were excluded. Only patients on clinical and biochemical remission were included in the study. However, this is a valid and interesting question and we have added it to discussion.
Reviewer 3 Report
Comments and Suggestions for Authors
1. Introduction: "Hepatitis B virus (HBV) is one of the major worldwide health challenges." There are no references to back up this statement. HBV causes public health problem partially due to its high genetic diversity, drug resistance, and immune-escape mutations in surface gene. The authors should introduce these aspectives. More references should be cited, with the following as an example (citing suggestion is optional):
Amino acid similarities and divergences in the small surface proteins of genotype C hepatitis B viruses between nucleos(t)ide analogue-naïve and lamivudine-treated patients with chronic hepatitis B. Antiviral Res. 2014 Feb;102:29-34. doi: 10.1016/j.antiviral.2013.11.015. Epub 2013 Dec 4. PMID: 24316031.
2. Table 1 should use "Control Group (5-ASA)" and "Study Group (IFX)" in the first row to be consistent with other tables.
3. Table 1 and 2 can merge as redundant info is presented.
4. Fig 2 should add statistical analysis results (P value).
5. Fig 2: 5-ASA group has two outliners with super high HBsAb level (250 IU/L). What caused the super high HBsAb in these two patients? Are they truely high positives or affected by other factors? Have the authors considered removing these two cases? If so, will removal of these two outliers affect the results and conclusions of this study?
6. Limitation of study should be discussed. This study lacks healthy normal control and has small case number of 5-ASA and IFX groups.
7. Even though IFX group has relatively lower HBSAb than 5-ASA group, the HBsAb level of IFX is still above 10 IU/L. What is the clinical significance of the findings of this study?
Comments on the Quality of English Languagenone
Author Response
- Introduction: "Hepatitis B virus (HBV) is one of the major worldwide health challenges." There are no references to back up this statement. HBV causes public health problem partially due to its high genetic diversity, drug resistance, and immune-escape mutations in surface gene. The authors should introduce these aspects. More references should be cited, with the following as an example (citing suggestion is optional):
Amino acid similarities and divergences in the small surface proteins of genotype C hepatitis B viruses between nucleos(t)ide analogue-naïve and lamivudine-treated patients with chronic hepatitis B. Antiviral Res. 2014 Feb;102:29-34. doi: 10.1016/j.antiviral.2013.11.015. Epub 2013 Dec 4. PMID: 24316031.
Thank you for your comment. We have added this statement to the introduction and cited it.
- Table 1 should use "Control Group (5-ASA)" and "Study Group (IFX)" in the first row to be consistent with other tables.
Thank you for your comment. We have modified as suggested.
- Table 1 and 2 can merge as redundant info is presented.
Thank you for your comment. We have modified as suggested and merged the tables.
- Fig 2 should add statistical analysis results (P value).
Thank you for your comment. We have added the p-value to figure 2.
- Fig 2: 5-ASA group has two outliners with super high HBsAb level (250 IU/L). What caused the super high HBsAb in these two patients? Are they truely high positives or affected by other factors? Have the authors considered removing these two cases? If so, will removal of these two outliers affect the results and conclusions of this study?
Thank you for your comment. We did not feel justified in removing the outliers because it can detrimentally affect the results of this study. However, we don’t have clear explanation for this possibly those patients may have received previous hepatitis B vaccination but they don’t recall doing so as those 2 patients were born in another country, we don’t have their vaccination record. This is added to the limitation section.
- Limitation of study should be discussed. This study lacks healthy normal control and has small case number of 5-ASA and IFX groups.
Thank you for your comment. We have added to the limitation as suggested.
- Even though IFX group has relatively lower HBSAb than 5-ASA group, the HBsAb level of IFX is still above 10 IU/L. What is the clinical significance of the findings of this study?
Thank you for your comment. Clinical significance is not currently know, however, patients on biologics are at higher risk of infections and tend to have faster decay in antibodies to vaccines overtime, this has been shown in patients with IBD on IFX after taking COVID-19 vaccine. We have added this point to discussion.
Kennedy NA, Goodhand JR, Bewshea C Contributors to the CLARITY IBD study, et al Anti-SARS-CoV-2 antibody responses are attenuated in patients with IBD treated with infliximab Gut 2021;70:865-875.
Round 2
Reviewer 2 Report
Comments and Suggestions for Authors
The authors changed their manuscript based on the reviewer's comments. Even though the main aspects of the study could not be improved, the presentation of the present study results is better now. The authors also addressed the comments regarding the previously vaccinated patients. The paper is acceptable now, but they may add more on the importance of their results for managing patients with IBD and biologic treatments.
Comments on the Quality of English LanguageSome editing is needed.
Author Response
Thank you for your comment and we appreciate your time spent on increasing the quality and clarity of our manuscript. We have added your suggestion to the discussion.
Reviewer 3 Report
Comments and Suggestions for Authors
The revisions have addressed all the concerns and questions of the reviewer.
Author Response
Thank you for your comment and we appreciate your time spent on increasing the quality and clarity of our manuscript.